# The Timing of Water and Beverage Consumption During the Day Among Children and Adults in the United States: Analyses of NHANES 2011–2016 Data

**DOI:** 10.3390/nu11112707

**Published:** 2019-11-08

**Authors:** Florent Vieux, Matthieu Maillot, Colin D. Rehm, Pamela Barrios, Adam Drewnowski

**Affiliations:** 1MS-Nutrition, 27 bld Jean Moulin Faculté de Médecine la Timone, Laboratoire C2VN, 13385 Marseille CEDEX 5, France; Florent.vieux@ms-nutrition.com (F.V.); mathieu.maillot@ms-nutrition.com (M.M.); 2Department of Epidemiology & Population Health, Albert Einstein College of Medicine, Montefiore Medical Center, New York, NY 10467, USA; colin.rehm@gmail.com; 3PepsiCo Inc, Purchase, NY 10577, USA; pamela.barrios@pepsico.com; 4Center for Public Health Nutrition, University of Washington, Box 353410, Seattle, WA 98195, USA

**Keywords:** water, beverages, National Health and Nutrition Examination Survey (NHANES 2011–2016), children, adults, meal, time of day, diurnal consumption, hydration

## Abstract

Dietary Guidelines for Americans 2015–20 recommend replacing sugar sweetened beverages (SSBs) with plain water in order to promote adequate hydration while reducing added sugar intake. This study explored how water intakes from water, beverages, and foods are distributed across the day. The dietary intake data for 7453 children (4–18 y) and 15,263 adults (>19 y) came from the National Health and Nutrition Examination Survey (NHANES 2011–2016). Water was categorized as tap or bottled. Beverages were assigned to 15 categories. Water intakes (in mL/d) from water, beverages, and food moisture showed significant differences by age group, meal occasion, and time of day. Plain water was consumed in the morning, mostly in the course of a morning snack and between 06:00 and 12:00. Milk and juices were consumed at breakfast whereas SSBs were mostly consumed at lunch, dinner, and in the afternoon. Children consumed milk and juices, mostly in the morning. Adults consumed coffee and tea in the morning, SSBs in the afternoon, and alcohol in the evening. Relatively little drinking water was consumed with lunch or after 21:00. Dietary strategies to replace caloric beverages with plain water need to build on existing drinking habits by age group and meal type.

## 1. Introduction

Dietary Guidelines for Americans 2015–2020 have recommended replacing sugar-sweetened beverages (SSBs) with plain drinking water [1]. The 2006 proposed guidance system for beverage consumption in the US also recommended choosing water over other beverages [2]. Analyses of 24 h dietary intakes from the most recent National Health and Nutrition Examination Survey (NHANES 2011–2016) suggest that these recommendations may have been effective. In recent years, plain drinking water, bottled and tap, has been replacing SSBs in the US diet [3]. The main sources of drinking water in the US have been tap water at home (288 mL/d), tap water away from home (301 mL/d), and bottled water from supermarkets and grocery stores (339 mL/d) [3]. Most SSBs have also come from stores. Stores contributed far more SSBs to the US diet than fast food restaurants, full service restaurants, and schools combined [4,5].

The 2015 Dietary Guidelines for Americans recommended a shift to reduce added sugar consumption to less than 10 percent of calories per day [1]. Among the suggested strategies were drinking SSBs less often, reducing SSB volume, or replacing SSBs with plain water on specific eating or drinking occasions [1]. Successful implementation of those strategies may require a better understanding of water and SSB consumption patterns during the day. The timing of and the frequency of drinking bouts and the amounts of fluids consumed can vary across population subgroups. Replacing SSBs with drinking water can also be challenging if the established SSB and water consumption patterns differ by age, race/ethnicity, or socioeconomic status (SES). 

For example, past analyses of NHANES data have shown a significant effect of age. Teenagers and young adults consumed the most fruit juices, SSBs, and water. Adults and older adults consumed much less SSBs but drank more coffee, tea, and alcohol [3]. Education and incomes also played a role. In past studies, lower-income groups consumed more regular soda; whereas higher-income groups tended to drink more diet soda [4,5]. Similarly, the consumption of whole milk was associated with lower SES; higher-income groups consumed more skim and reduced-fat milk [4,5]. 

A socio-economic gradient was recently observed for the consumption of tap water [3]. Analyses of NHANES 2011–2016 data showed, for the first time, that most tap water was consumed by groups of higher education and incomes [3]. This may be the result of powerful new marketing campaigns that hope to change the way that Americans think about water, bottled and tap [6]. The newly observed social gradient may also be a direct result of the “Flint effect” and the growing distrust of municipal water systems in low-income areas and among communities of color [7,8].

Aligning daily beverage choices with healthy eating patterns is a key component of many dietary intervention programs [1,9,10]. However, such dietary strategies may need to build on existing beverage consumption patterns and the timing of water and beverage consumption during the day. Here, the available data are limited. Only a few studies on children in the UK and in France have examined water and beverage consumption patterns by meal and time of day [11,12]. Earlier US based studies have examined sourcing locations but not by meal type or time of day [4,5]. 

The timing of beverage consumption in the course of the day may have additional implications for adequate hydration. There is an emerging mythology about the correct time to drink water in the course of the day. One strategy is to drink water 30 minutes before a meal, during a meal. and after a meal, but no more [13]. Another is to drink water early in the morning, soon after waking up [14]. Additional recommendations are to drink water before, during, and after a workout, before a bath, and just before going to bed at night. Drinking water at the correct time is alleged to help prevent stomach pain, irritable bowel syndrome, fatigue, overeating, high blood pressure, and even heart attack and stroke [15]. However, evidence in support of those strategies is limited.

One recent suggestion was that mild dehydration may occur in a transient manner when water and fluids are not consumed, either because of poor access to water or beverages or because of poor drinking and eating habits [16]. The present study explored daily fluctuations in water intakes from water, beverages, and foods in a large and nationally representative sample of children and adults in the US.

## 2. Materials and Methods 

### 2.1. Dietary Intake Databases

Consumption data for drinking water, beverages, and foods came from 3 cycles of the nationally representative National Health and Nutrition Examination Surveys (NHANES), corresponding to years 2011–2012, 2013–2014, and 2015–2016 [17]. The three NHANES cycles provided a nationally representative sample of 7453 children (aged 4–18 y) and 15,263 adults (aged ≥19 y).

The NHANES 24-hour recall uses a multi-pass method, conducted by a trained interviewer using a computerized interface. Respondents report the types and amounts of all food and beverages consumed in the preceding 24 hours, from midnight to midnight [18,19]. Respondents first identify a quick list of foods and beverages, reporting both meal occasion and time of day. A more detailed cycle then records the amounts consumed, followed by a final probe for any often-forgotten foods. Day 1 interviews are conducted by trained dietary interviewers in a mobile examination center. Day 2 interviews are conducted by telephone some days later [19]. 

For children 4–5 y, dietary recall is completed entirely by a proxy respondent (i.e., a parent or guardian with knowledge of the child’s diet) [19]. Children 6–11 y are primary respondents, but a proxy respondent is present and able to assist. Children 12–19 y are primary respondents but can be assisted by an adult who has knowledge of their diet [19]. We used a combination of the 1-day value and the 2-day mean to make use of all available dietary data. This method included all NHANES participants, even those without a second recall. 

### 2.2. Participant Characteristics

NHANES participants were stratified by sex and age. The age group cut-points were: 4–8, 9–13, 14–18, 19–30, 31–50, 51–70, and >70 y. These age groups generally correspond to the age groups used by the IOM. Race/ethnicity was defined as non-Hispanic white, non-Hispanic black, Mexican American, Other Hispanic, and other/mixed race. Family income-to-poverty ratio (IPR) is an index of socioeconomic status; the cut-points for the IPR were <1, 1–1.99, 2–3.49, and ≥3.5.

### 2.3. Water and Beverage Categories

Plain drinking water was split into tap and bottled. Beverages were classified into 15 categories: milk and milk beverages, milk substitutes (soy milk), citrus juices, non-citrus juices, diet soda, regular soda, ready-to-drink tea, ready-to-drink (RTD) coffee, fruit drinks, sports drinks, energy drinks, hot tea/coffee, alcoholic beverages, flavored, carbonated or enhanced water, and supplemental beverages. The present analyses of water intakes from beverages were for beverages only; for example, milk consumed with cereal (i.e., not as a beverage) was counted in the food category. The USDA Food and Nutrient Database for Dietary Studies (FNDDS), used to establish energy and nutrient content of individual diets, has been revised in parallel to each NHANES cycle [20].

The NHANES 24-hour recall for each participant provides information on the amount in grams of each food and beverage consumed. The present results were for mL of water content from selected beverages, and not for the volume of the beverages themselves (which may not be 100% water). Moisture from foods was calculated as well. 

### 2.4. Water and Beverages by Meal Type and Time of Day

Eating occasions were classified as follows: breakfast, morning snack, lunch, afternoon snack, dinner, and evening snacks. That information was obtained by self-reporting. The distribution of water and beverage intakes was also captured by time of day. The 24 h temporal profile of water and beverage consumption was framed in 3 h intervals, starting at 06:00 (6 a.m.) until midnight. The time intervals were 06:00–09:00; 09:00–12:00; 12:00–15:00, 15:00–18:00, 18:00–21:00, 21:00–00:00, and 00:00–06:00.

### 2.5. Data Availability and Ethical Approval

The necessary IRB approval for NHANES was obtained by the National Center for Health Statistics (NCHS) [21]. Adult participants provided written informed consent. Parental/guardian written informed consent was obtained for children. Children/adolescents ≥12 y provided additional written consent. All NHANES data are publicly available on the NCHS and USDA websites [17]. Per University of Washington (UW) policies, public data that do not involve “human subjects” and their use requires neither IRB review nor an exempt determination. Such data may be used without any involvement of the Human Subjects Division or the UW Institutional Review Board.

### 2.6. Statistical Analyses

The survey-weighted mean intakes of total water were evaluated overall and by age group, sex, race/ethnicity, and family income-to-poverty ratio. All analyses accounted for the complex survey design of NHANES and reflected the dietary behaviors of the US adult population from 2011 to 2016. 

The consumption of water and beverages was evaluated for the entire population and for population sub-groups. Survey-weighted means and corresponding standard errors were obtained. All analyses were conducted using SAS software, version 9.4 (SAS Institute Inc., Cary NC, USA) by using SURVEYREG, SURVEYMEANS, and SURVEYFREQ procedures.

## 3. Results

### 3.1. Total Water Intakes from Water and Beverages

Table 1 shows total water intakes from water, beverages and foods in mL/d by sex, eating occasion, and time of day. Total water intake was 2718 mL/d, of which 2100 mL/d (77%) came from water and beverages and 618 mL/d (23%) came from food moisture. Drinking water provided 1066 mL/d, and caloric and non-caloric beverages provided 1034 mL/d of water. Most drinking water came from the tap (tap: 661 mL/d; bottled 404 mL/d). The dietary sources of water were beverages (38%), tap water (24%), bottled water (14%), and food moisture (23%).

Men consumed more total water and more beverages than did women; there was no sex effect for water consumption. Non-Hispanic Whites consumed the most water, the most beverages and the most total water; lowest water consumers were Non-Hispanic Blacks. Mexican Americans drank the most bottled water. Water and beverage consumption also increased with the income to poverty ratio (IPR). Income effects were observed for water and beverages, the effect for tap water was particularly strong (496 vs. 821 mL/d). 

Figure 1A (top panel) provides a visual representation of these consumption patterns. Dinner and lunch followed by breakfast were the peak times for water consumption from food moisture. Beverages were consumed throughout the day with peaks at dinner, breakfast, lunch, and the morning snack. Tap water was associated mostly with the morning snack and so was bottled water. Most water from tap and bottled water was consumed in the course of the morning snack. The least tap and bottled water was consumed at breakfast and lunch. 

Figure 1B (bottom panel) shows the corresponding water consumption patterns by time of day. Tap and bottled water were mostly consumed between 06:00 and 12:00. Water consumption dropped by half in the afternoon and evening. Beverages were mostly consumed between 06:00 and 12:00 and again between 18:00 and 21:00. Peaks for water from food moisture were at 18:00 and 21:00 (dinner) and 12:00 to 15:00 (lunch). Breakfast and snacks did not provide substantial food moisture.

### 3.2. Water and Beverage Consumption by Time of Day

Figure 2A (top panel) shows water intakes from water and beverages for all ages (>4 y) by meal occasion. Most water, tap and bottled, was consumed during the eating occasion identified as the morning snack. Smaller amounts were consumed during the afternoon snack. Most SSBs were consumed with lunch and dinner; consumption of SSBs during breakfast and morning snack was low. Relatively little water was consumed at lunch.

Figure 2B (bottom panel) shows water intakes from water and beverages for all ages (>4 y) by time of day. Tap and bottled water were mostly consumed between 06:00 and 12:00. Consistent with expectations, brewed tea and coffee were consumed between 06:00 and 12:00, times corresponding to breakfast and the morning snack. Regular and diet sodas were mostly consumed between 12:00 and 21:00. Alcohol was consumed in the afternoon and in the evening. Coffee and tea were less likely to be consumed at dinner and in the evening, as compared to morning. Beverage consumption dropped sharply after 21:00.

### 3.3. Water and Beverage Consumption by Age Group and Time of Day

Figure 3, Figure 4, Figure 5, Figure 6, Figure 7, Figure 8 and Figure 9 show the time of consumption of water from water and beverages, separately for each age group. The data showed that different age groups have very specific patterns of water and beverage consumption depending on the time of day. The dataset is in Appendix A. 

## 4. Discussion

The present results are among the first to document the timing of water and beverage intakes around the clock in a large and representative NHANES 2011–2016 sample of US children and adults. The present results have important implications for the promotion of healthy beverage choices, notably the ongoing attempts to replace SSBs with plain drinking water.

There is very little science on population water consumption patterns during the day. One recent study, conducted in Greece, explored the fluctuation in water intakes and hydration indices during the day, looking for signs of transient dehydration in a sample of healthy adults [16]. While water intakes did go up and down during the day, as they did here, the term fluctuation generally refers to an unpredictable and irregular rising and falling. As the present results show, the timing of water and beverage consumption followed predictable patterns. SSBs were consumed with lunch and dinner and in the afternoon but rarely at breakfast or the mid-morning snack. Water was consumed largely in the morning and rarely at night. Adults drank coffee in the morning and alcohol in the evening [22]. Beverage choices and consumption patterns varied with age. Whereas children aged 4–18 y consumed water between 18:00 and 21:00, adults were more likely to consume less water and more alcohol in the same time slot. Furthermore, while children consumed milk in the morning, adults tended to drink tea or coffee during this time.

The present data add to past work on the impact of caloric beverages consumed separately or with a meal on total energy intakes. In experimental studies, when caloric beverages were presented shortly before or with a pizza meal, no energy compensation was observed. Caloric beverages consumed freely at meal times added calories to the meal [23,24]. By contrast, other studies showed that the presentation of a stand-alone liquid preload reduced energy intakes at the test meal; however, the effects were sometimes inconsistent and full energy compensation was rarely observed [25]. 

Excessive SSB consumption is thought to contribute to childhood obesity [26]. Dietary Guidelines for Americans have stressed the importance of healthier beverage choices to be made throughout the day [1]. Ensuring access to safe, free drinking water in schools is an important CDC initiative that is intended to increase water consumption, help maintain hydration and reduce energy intake when substituted for SSBs [27]. School-based strategies to replace SSBs with plain drinking water have ranged from limiting sales in cafeterias, vending machines, and competitive food outlets to featuring teachers as competitive role models [9]. 

The present analyses support the CDC initiative but for a different reason. The CDC report notes that more than 95% of children are enrolled in schools and typically spend 6 h at school each day. We note that those times are in the morning (typically), which are the peak times for water consumption. There seems to be less competition during the morning snack from other beverages. Another productive strategy would be to promote water consumption with the school lunch meal.

We were surprised to see that water did not figure prominently in the afternoon snack. Though the total amount of water consumed was comparable to that during the morning snack, those beverages were SSBs, tea, and alcohol (for adults). Promoting water consumption by children in the afternoon may be a potential intervention strategy. 

By contrast, promoting more water consumption in the morning might be a viable strategy for adults. There are opportunities to increase water consumption at lunch. By contrast, water is unlikely to displace morning coffee, especially with older adults or the afternoon tea [22].

The social gradient in water consumption has been addressed before [28]. Plain drinking water, bottled and tap, accounts for 38% of daily water intake from all sources including food moisture. The intake is slightly lower for lower income and minority populations. Hispanic Americans drink more bottled water and tap water but other minorities do not [3]. One issue is public trust in the municipal water system—the provision of safe water in schools and community settings is critical to the adoption of healthier beverage choices [27,28].

This study had limitations. First, the NHANES 2011–2016 data are based on dietary self-reports, still the default practice in large population-based studies. Second, the within-day variations by meal and time interval were close but not quite the same. For example, the consumption of water was high between 06:00 and 12:00, but it was associated not with breakfast but with a morning snack. People who reported consuming a morning snack did not necessarily have breakfast. Finally, the NHANES 2011–2016 data are cross sectional, not allowing for causal inferences to be made. The potential impact on water consumption patterns on health outcomes of interest cannot be determined.

## 5. Conclusions

Present analyses of diurnal fluctuations in water intakes can inform dietary strategies for maintaining adequate hydration while reducing the consumption of added sugars. The most effective strategies for behavioral change are those that build on existing habits and consumption patterns [1].

## Figures and Tables

**Figure 1 nutrients-11-02707-f001:**
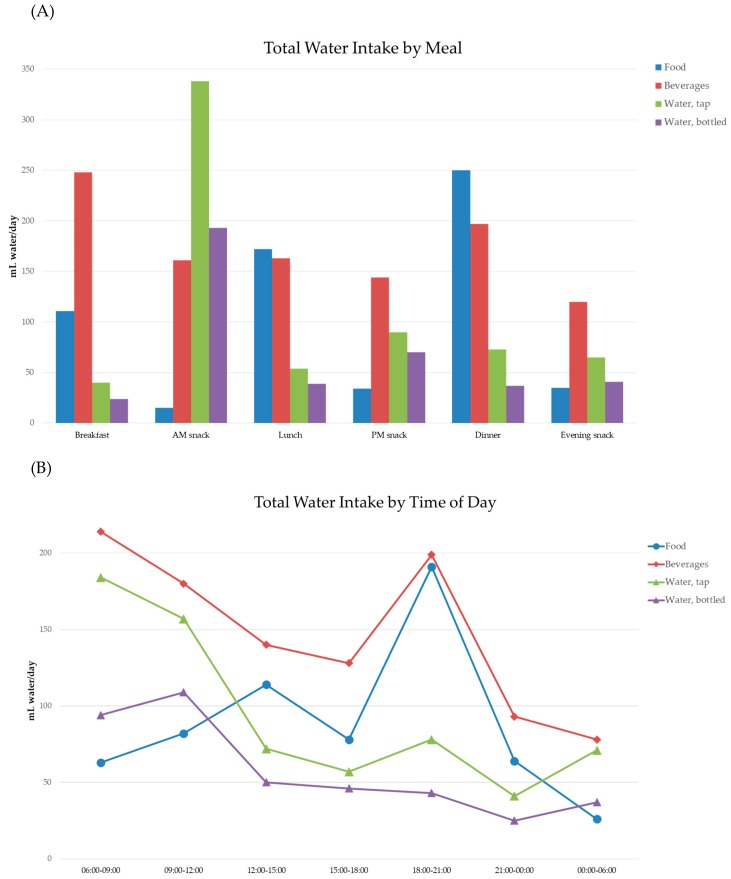
(**A**) (top panel) Total water intake from water, beverages, and foods by eating occasion. (**B**) (bottom panel) Total water intake from water, beverages, and foods by time of day. Data are for participants >4 y.

**Figure 2 nutrients-11-02707-f002:**
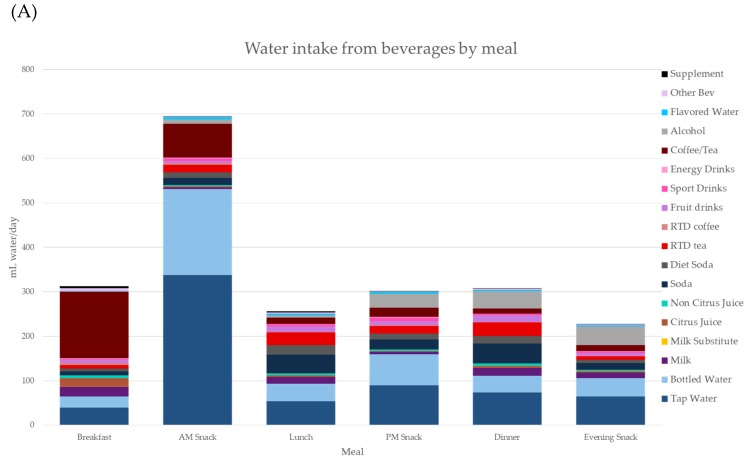
(**A**) (top panel) Water intakes from water and beverage categories by eating occasion. (**B**) (bottom panel) Water intakes from water and beverage categories by time of day. Data are for participants >4 y.

**Figure 3 nutrients-11-02707-f003:**
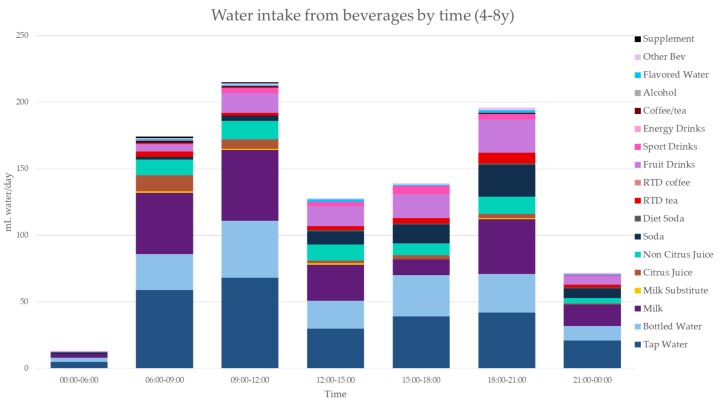
Water intakes (mL/d) by beverage type and time of day for the 4–8 y age group (NHANES 2011–2016).

**Figure 4 nutrients-11-02707-f004:**
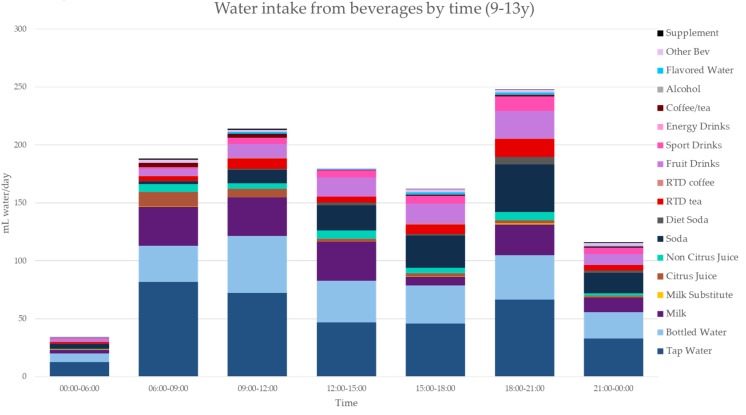
Water intakes (mL/d) by beverage type and time of day for the 9–13 y age group (NHANES 2011–2016).

**Figure 5 nutrients-11-02707-f005:**
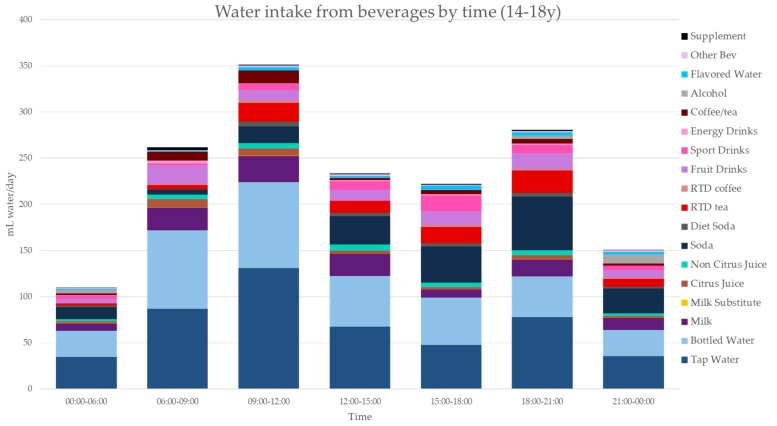
Water intakes (mL/d) by beverage type and time of day for the 14–18 y age group (NHANES 2011–2016).

**Figure 6 nutrients-11-02707-f006:**
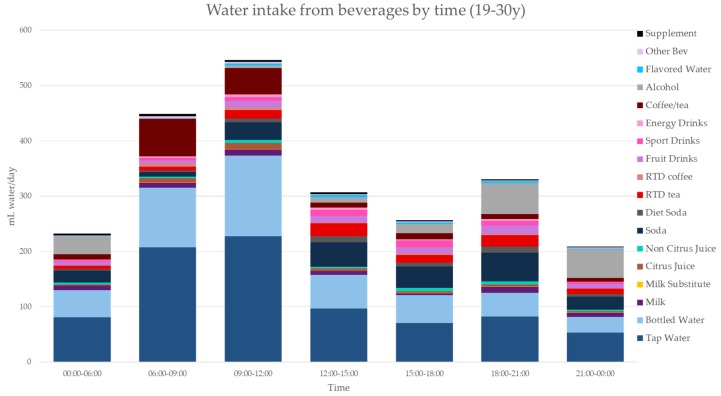
Water intakes (mL/d) by beverage type and time of day for the 19–30 y age group (NHANES 2011–2016).

**Figure 7 nutrients-11-02707-f007:**
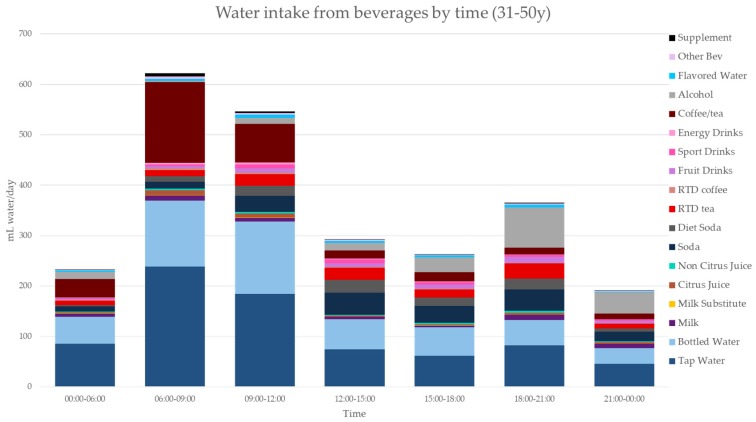
Water intakes (mL/d) by beverage type and time of day for the 31–50 y age group (NHANES 2011–2016).

**Figure 8 nutrients-11-02707-f008:**
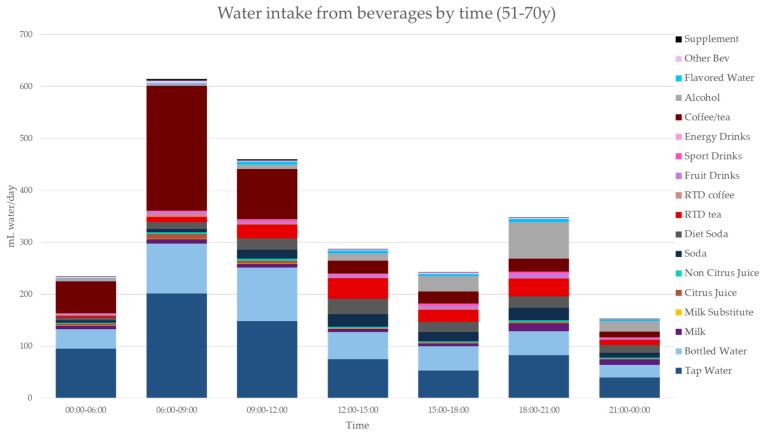
Water intakes (mL/d) by beverage type and time of day for the 51–70 y age group (NHANES 2011–2016).

**Figure 9 nutrients-11-02707-f009:**
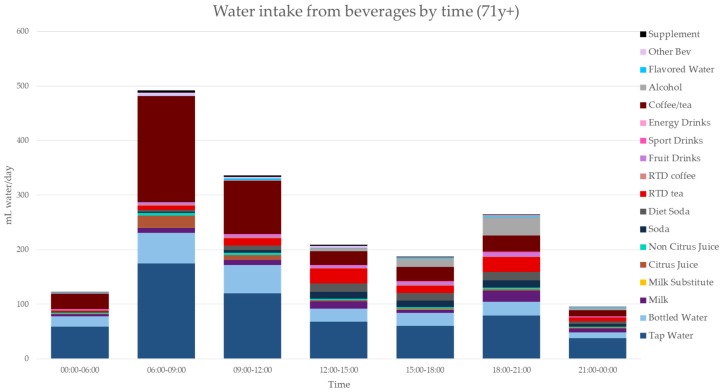
Water intakes (mL/d) by beverage type and time of day for the 71+ y age group (NHANES 2011–2016).

**Table 1 nutrients-11-02707-t001:** Water intakes (mL/d) from beverages and foods (mean, standard error) by individual characteristics and time. Data are pooled for 2011–2016 NHANES cycles. Data are means and SEMs.

	Water, Beverages, and Foods (mL/d)			Water, Tap and Bottled (mL/d)			Beverages (mL/d)
	Total mL/d	Water and Beverages mL/d	Food Moisture mL/d	Water, Tap and Bottled	Tap Water	Bottled Water	Beverages
All >4y (*N* = 22,716)	2718(27)	2100(26)	618(6)	1066(20)	661(24)	404(16)	1034(17)
Sex							
Men (*N* = 11,206)	2949(36)	2285(34)	664(6)	1071(24)	676(25)	395(16)	1214(24)
Women (*N* = 11,510)	2495(25)	1921(24)	574(7)	1060(20)	647(27)	414(17)	861(14)
*p*-value for effect	<0.01	<0.01	<0.01	0.59	0.14	0.10	<0.01
Race/ethnicity							
NH White (*N* = 7802)	2879(31)	2266(29)	613(7)	1109(26)	781(27)	328(16)	1157(23)
NH Black (*N* = 5365)	2249(36)	1695(32)	554(7)	894(29)	359(23)	534(23)	802(14)
Mexican American (*N* = 4209)	2487(51)	1857(48)	630(10)	997(36)	391(21)	605(26)	861(19)
Other Hispanic (*N* = 2473)	2505(38)	1897(34)	609(8)	1041(33)	497(41)	544(34)	856(18)
Other (*N* = 3378)	2635(38)	1899(38)	737(13)	1092(26)	658(31)	434(29)	807(24)
*p*-value for effect	<0.01	<<0.01	<0.01	<0.01	<0.01	<0.01	<0.01
IPR							
<1 (*N* = 5633)	2461(49)	1888(48)	573(6)	932(37)	496(45)	436(21)	956(30)
1–1.99(*N* = 5545)	2579(38)	1989(36)	590(8)	1003(27)	555(27)	448(22)	986(27)
2–3.49 (*N* = 4209)	2692(40)	2092(39)	601(9)	1036(28)	643(28)	394(21)	1055(25)
>3.49 (*N* = 5558)	2952(38)	2288(37)	664(9)	1181(29)	821(32)	360(20)	1108(19)
Missing (*N* = 1771)	2616(56)	1989(52)	627(13)	1074(49)	609(44)	464(29)	916(27)
p-value for effect	<0.01	<0.01	<0.01	<0.01	<0.01	<0.01	<0.01
Type of meal							
Breakfast	425(5)	314(4)	111(1)	65(2)	40(2)	25(1)	249(4)
AM snack	707(20)	692(19)	15(1)	531(17)	338(16)	193(10)	161(7)
Lunch	428(5)	256(4)	172(3)	93(2)	54(2)	39(2)	163(3)
PM snack	337(7)	303(7)	34(1)	159(4)	90(4)	70(2)	144(5)
Dinner	559(4)	308(3)	250(3)	111(3)	74(3)	37(2)	197(3)
Evening snack	262(4)	227(4)	35(1)	107(3)	65(3)	41(2)	120(3)
*p*-value for effect	<0.01	<0.01	<0.01	<0.01	<0.01	<0.01	<0.01
Time of consumption (h)							
06:00–09:00	555(12)	492(12)	63(1)	278(9)	184(9)	94(6)	214(6)
09:00–12:00	528(9)	446(9)	82(1)	266(8)	157(7)	109(5)	180(4)
12:00–15:00	377(5)	263(5)	114(2)	123(4)	72(3)	50(2)	141(4)
15:00–18:00	310(4)	232(4)	78(1)	104(2)	57(2)	46(2)	128(3)
18:00–21:00	512(6)	320(5)	191(3)	121(3)	78(4)	43(2)	200(4)
21:00–00:00	224(4)	160(3)	64(2)	66(2)	42(2)	25(1)	93(3)
00:00–06:00	213(6)	186(6)	26(1)	108(5)	71(5)	37(3)	78(3)
*p*-value for effect	<0.01	<0.01	<0.01	<0.01	<0.01	<0.01	<0.01

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
