# Peer review of "The Timing of Water and Beverage Consumption During the Day Among Children and Adults in the United States: Analyses of NHANES 2011–2016 Data"

_nutrients, 2019, doi:10.3390/nu11112707_

Round 1
Reviewer 1 Report
This is an interesting study examining the timing of water and beverage consumption across different ages ranges among children and adults; however, it is not acceptable in its present form as there is a lack of detail in some sections. In order to provide more flow and clarity to the paper, please address the following points.
In the introduction, there was no mention of the Beverage Guidance Panel which is used to guide beverage consumption in the US. It recommends water over other beverages. Please refer to this. Lines 87-90: A) There is mention of race/ethnicity and family IPR in the participant characteristics section. This is also mentioned again in line 123 in the statistical analyses. However, there is no mention again of these two factors nor have any results been shown with respect to these two parameters. Please include these in the results section. Graphs or tables with these results should accompany these results. B) Should your results be adjusted for these two factors? Please provide a rationale as to why or why not? Lines 91-97: Tap and bottled water were included. Was there a difference between still and carbonated water and why were these not included? This is important to consider since the focus of your paper is on water. Lines 91-97: With respect to plant-based beverages such as soy, why were other beverages such as rice or almond beverages not included? Perhaps a category on plant-based beverages would have been more appropriate? Please clarify. The word gender should be changed for ``sex``. Figures should read Figure 1A and 1B rather than Figure 1top and Figure 1bottom. Line 156: The title reads water and beverage consumption by age group and time of day, but it should really read water and beverage consumption by meal occasion and time of day. I would then include a separate section on water and beverage consumption by age group and include your supplementary graphs there. Figure 1: In the legend there is mention of ``other`` and ``enhanced water``, but there is no description to describe what these are. Please clarify. Figure 1: Please describe RTD. This may also be clarified in the methods section in order to clarify the abbreviation. Discussion: This section requires a lot of work and would benefit from a bit more interpretation of the results and what they mean. It would also be more clear to restructure the discussion. This would provide more clarity and flow for the reader. As written, it is a bit difficult to follow. For example, the first paragraph (lines 176-182) would be more appropriate in the introduction that in the discussion. I would start the discussion with the main findings (e.g. lines 187-191) and then continue from there. Please provide more detail for the result presented on line 191. Line 188: It is unclear what is meant by the timing was regular and precise. Please clarify. Lines 194: How does one arrive to this conclusion from these results? What about the importance of timing? Is it important? Is it better to have water before meals or with meals? You may also take a look at short-term studies on beverage consumption with and without meals by Panahi et al. (Appetite 2013a and 2013b) as reference. Does the time of day or meal occasion matter with respect to water intake physiologically? If water safety is an issue, is this resulting in individuals not consuming enough water? This point was mentioned several time, but because ethnicity and IPR results were not shown it is unclear why this is important for this paper. Please simplify the conclusion. It should also match that of the one in the abstract. I would include a table with participant characteristics. Please review the language. There are some minor changes to be made.Author Response
Please see attachment

Reviewer 2 Report
The authors appear to have done a thorough job analyzing NHANES dietary recall data to try to better understand when beverages are most frequently consumed during the day. However, a key limitation for me as a reviewer is that I am struggling to understand what the purpose of this study was meant to be. The introduction runs through some existing evidence on consumption of different beverages in different populations, and then makes a claim that we need to know about the time of day during which beverages were consumed for interventions - but this claim has no citations, and I don't think the authors really justify why this is necessary. Is there evidence that existing intervention studies have been unsuccessful because they don't take daily consumption patterns into account? I'm really not aware of any, and I'm also not aware of evidence in general that suggests that it's necessary to understand what time of day people consume different things in order to try to maximize their intake during that time. Then, the first paragraph of the Discussion section seems to imply that the impetus for the study was to debunk myths about the timing of water consumption, but this is not mentioned anywhere else. I think a stronger justification for why this study is useful is warranted, as well as a clear statement of study aims and hypotheses.
Other comments:
Line 70 - these sample sizes sound small for three waves of NHANES - is this after exclusions for those with dietary data? Were there exclusions based on missing demographic variables?
Line 79- age groupings are not mutually exclusive
Lines 93-95 - why utilize these beverage groupings? Is there a citation for this?
Lines 101-103 - why was only the mLs of water for each beverage analyzed, rather than just the total volume of the beverages themselves? I could sort of see this being a way to analyze if the goal of the study was to estimate how total water intake is distributed across foods and beverages across the day, but it doesn't seem from elsewhere that this is the study aim.
Line 103 - how was moisture from food calculated?
Lines 106 - 110 - why were the data analyzed both by mealtimes and by these three hour intervals - what was the motivation for doing both of these? Not sure both are necessary.
Table 1- I assume the numbers in parentheses are standard errors? But not clearly labeled.
lines 187-188 - Stating that the "timing of water and beverage consumption was, in fact, regular and precise" is outside the scope of what is possible to state with this analysis. The authors did not test how regular individual study participants' consumption patterns were from day to day, nor did they test how much participants' individual patterns deviated from the population mean patterns. I am unsure what the authors mean by this.
Lines 232-233 - this statement needs a citation.
